# Study on NGF and VEGF during the Equine Perinatal Period—Part 2: Foals Affected by Neonatal Encephalopathy

**DOI:** 10.3390/vetsci9090459

**Published:** 2022-08-26

**Authors:** Nicola Ellero, Aliai Lanci, Vito Antonio Baldassarro, Giuseppe Alastra, Jole Mariella, Maura Cescatti, Carolina Castagnetti, Luciana Giardino

**Affiliations:** 1Department of Veterinary Medical Sciences (DIMEVET), University of Bologna, 40064 Bologna, Italy; 2Health Science and Technologies Interdepartmental Center for Industrial Research (HST-ICIR), University of Bologna, 40064 Bologna, Italy; 3IRET Foundation, 40064 Bologna, Italy

**Keywords:** placental insufficiency, dystocia, Neonatal Encephalopathy, nerve growth factor, vascular endothelial growth factor, brain derived neurotrophic factor, thyroid hormones

## Abstract

**Simple Summary:**

Based on human medicine, Neonatal Encephalopathy is the term used by equine clinicians for newborn foals which develop a variety of non-infectious neurological signs in the immediate postpartum period. It has become the preferred term because it does not imply a specific underlying etiology or pathophysiology, as hypoxia and ischemia may not be recognized in all cases. Understanding the underlying pathophysiology is important in formulating a rational approach to diagnosis. Our aim is to clinically characterize a population of foals spontaneously affected by Neonatal Encephalopathy and to evaluate the levels of trophic factors, such as nerve growth factor and vascular epithelial growth factor, and thyroid hormones obtained at birth/admission from a population of affected foals and in the first 72 h of life/hospitalization, as well as the expression of trophic factors in the placenta of mares that delivered foals affected by Neonatal Encephalopathy. The less pronounced decrease of the two trophic factors compared to healthy foals, their close relationship with thyroid hormones over time, and the dysregulation of trophic factor expression in placental tissues, could be key regulators in the mechanisms of equine Neonatal Encephalopathy.

**Abstract:**

Neonatal Encephalopathy (NE) may be caused by hypoxic ischemic insults or inflammatory insults and modified by innate protective or excitatory mechanisms. Understanding the underlying pathophysiology is important in formulating a rational approach to diagnosis. The preliminary aim was to clinically characterize a population of foals spontaneously affected by NE. The study aimed to: (i) evaluate nerve growth factor (NGF) and vascular endothelial growth factor (VEGF) levels in plasma samples obtained in the affected population at parturition from the mare’s jugular vein, umbilical cord vein and foal’s jugular vein, as well as in amniotic fluid; (ii) evaluate the NGF and VEGF content in the plasma of foals affected by NE during the first 72 h of life/hospitalization; (iii) evaluate NGF and VEGF levels at birth/admission in relation to selected mare’s and foal’s clinical parameters; (iv) evaluate the relationship between the two trophic factors and thyroid hormone levels (TT3 and TT4) in the first 72 h of life/hospitalization; and (v) assess the mRNA expression of NGF, VEGF and brain-derived neurotrophic factor (BDNF), and their cell surface receptors, in the placenta of mares that delivered foals affected by NE. Thirteen affected foals born from mares hospitalized for *peripartum* monitoring (group NE) and twenty affected foals hospitalized after birth (group exNE) were included in the study. Dosage of NGF and VEGF levels was performed using commercial ELISA kits, whereas NGF, VEGF, and BDNF placental gene expression was performed using a semi-quantitative real-time PCR. In group NE, NGF levels decreased significantly from T0 to T24 (*p* = 0.0447) and VEGF levels decreased significantly from T0 to T72 (*p* = 0.0234), whereas in group exNE, only NGF levels decreased significantly from T0 to T24 (*p* = 0.0304). Compared to healthy foals, a significant reduction of TT3 levels was observed in both NE (T24, *p* = 0.0066; T72 *p* = 0.0003) and exNE (T0, *p* = 0.0082; T24, *p* < 0.0001; T72, *p* < 0.0001) groups, whereas a significant reduction of TT4 levels was observed only in exNE group (T0, *p* = 0.0003; T24, *p* = 0.0010; T72, *p* = 0.0110). In group NE, NGF levels were positively correlated with both TT3 (*p* = 0.0475; r = 0.3424) and TT4 levels (*p* = 0.0063; r = 0.4589). In the placenta, a reduced expression of NGF in the allantois (*p* = 0.0033) and a reduced expression of BDNF in the amnion (*p* = 0.0498) were observed. The less pronounced decrease of the two trophic factors compared to healthy foals, their relationship with thyroid hormones over time, and the reduced expression of NGF and BDNF in placental tissues of mares that delivered affected foals, could be key regulators in the mechanisms of equine NE.

## 1. Introduction

Based on human medicine, Neonatal Encephalopathy (NE) is the term used by equine clinicians for newborn foals which develop a variety of non-infectious neurological signs in the immediate postpartum period. So far, the terms used included neonatal maladjustment syndrome, hypoxic-ischemic encephalopathy, hypoxic-ischemic syndrome, perinatal asphyxia syndrome, neonatal maladaptation syndrome, barkers, wanderers, convulsives, and dummies [1,2,3,4,5,6,7,8,9,10,11], but NE has become the preferred term because it does not imply a specific underlying etiology or pathophysiology [12], as hypoxia and ischemia may not be recognized in all cases.

The assessment of brain injury in newborn foals affected by NE currently relies on clinical examination. The disease in foals is recognized by many behavioral abnormalities and other signs of neurologic dysfunction including alterations in respiratory function, changes in muscle tone, changes in responsiveness, vestibular signs, and autonomic disturbances [13]. However, these tools are limited by their subjective nature and the expertise required for interpretation, and peripheral blood biomarkers which reflect end-organ injury may provide objective quantitative measures that are free from these limitations [10]. Nevertheless, there are no blood-based biomarkers in current clinical use for foals with NE. Unlike in the human infant, the prognosis of foals with a single diagnosis of NE, in terms of short- and long-term neurological damage, is good [12].

Overall, equine NE can be the consequence of adverse peripartum events leading to ischemia/hypoxia/inflammation in the prepartum period (e.g., high-risk pregnancy, mares’ systemic illness, placental insufficiency) and at parturition (e.g., premature placenta separation, dystocia, cesarian section) [12]. Causes of high-risk pregnancy in the equine species can be of maternal, fetal, or placental origin, and most placental conditions pose limited risks to the mare, but significant risks to the fetus [14]. Placental insufficiency is an ill-defined condition that has been identified in a large retrospective study as responsible for more than 60% of pregnancy losses due to abortion, stillbirth, and neonatal death [14,15]. Several factors contribute to placental insufficiency such as premature placental separation, placental villous hypoplasia, placental thickening, and placentitis [16,17,18].

Neurotrophins, such as nerve growth factor (NGF) and brain-derived neurotrophic factor (BDNF), are known to play an important role in pre- and post-natal brain development, making them of great interest for the diagnosis and treatment of numerous neurological disorders [19]. Vascular endothelial growth factor (VEGF) has also been implicated in several acute neurological disorders, such as in brain ischemia, where it can mediate positive effects according to three different mechanisms. VEGF can stimulate angiogenesis and modulate vascular permeability, exert direct neuroprotective effects, or promote neurogenesis [20]. In addition to its neuroprotective role, VEGF has also negatively been implicated in blood–brain barrier break-down after ischemia and in mediating inflammatory responses [21,22]. Recently, the authors measured NGF and VEGF levels in the plasma of mares diagnosed with normal pregnancy and parturition, in the amniotic fluid, in the umbilical cord vein, and in the plasma of their healthy foals in the first 72 h of life [23].

Reported laboratory abnormalities in foals affected by NE include low thyroid hormone concentrations [7]. Although not diagnostic, thyroid disfunction is a frequent condition in critically ill foals [24], and low thyroid hormones are associated with disease severity also in sepsis [25] and prematurity [26]. Although a possible interaction of thyroid hormones with members of the neurotrophin family has been investigated in the equine species under physiological conditions [23], to the authors’ knowledge, the relationship between trophic factors and thyroid hormones in the equine NE is still under investigation.

As the NGF and VEGF levels at parturition and in the early stages of equine neonatal life under physiological conditions have been elucidated in the first part of the study [23], the role of the two trophic factors can now be investigated in a population of foals affected by NE. The preliminary aim of the present investigation, as a pilot study, was to clinically characterize a population of foals spontaneously affected by NE. Particularly (i) to evaluate NGF and VEGF levels in plasma samples obtained in the affected population at parturition from the mare’s jugular vein, umbilical cord vein and foal’s jugular vein, as well as in amniotic fluid; (ii) to evaluate NGF and VEGF content in the plasma of foals affected by NE during the first 72 h of life/hospitalization; (iii) to evaluate NGF and VEGF levels at birth/admission in relation to selected mare’s and foal’s clinical parameters; (iv) to evaluate the relationship between the two trophic factors and thyroid hormone levels (TT3 and TT4) in the first 72 h of life/hospitalization; and (v) to assess the mRNA expression of NGF, BDNF, and VEGF and their cell surface receptors in the placenta of mares that delivered foals affected by NE. This study was based on the hypothesis that circulating NGF and VEGF levels, which theoretically reflect central levels and then cerebral status, should differ between healthy neonatal foals and foals affected by NE, and that trophic factors also play a key role in metabolic activities, cell differentiation and development through their interaction with the hypothalamus–pituitary–thyroid (HPT) axis.

## 2. Materials and Methods

### 2.1. Population

Thirteen sick foals born from mares hospitalized for *peripartum* monitoring and twenty sick neonatal foals hospitalized within 24 h after birth at the Perinatology and Reproduction Unit (Equine Clinical Service, Department of Veterinary Medical Sciences) of the University of Bologna during 2018–2021 foaling seasons were included in Part 2 of the study. The population of 14 healthy foals born from mares with normal pregnancy and parturition hospitalized for *peripartum* monitoring examined in Part 1 of the study [23] represented the healthy control group (group H).

Upon a pregnant mare’s admission, a complete clinical evaluation, including blood count (ADVIA 2120 analyzer, Siemens Healthcare srl, Milan, Italy) and transrectal ultrasonography were performed. Based on the judgement of the clinician, transabdominal ultrasonography was performed when indicated. Subsequently, mares were clinically evaluated twice a day and by ultrasonography every 5–10 days until parturition. When an increase in the combined thickness of the uterus and placenta (CTUP) was observed, a cervical swab was performed to obtain a bacterial culture and a focused treatment was eventually started based on the clinician’s judgment. After delivery, a macroscopic and histopathological examination of the placenta was performed in all mares.

High-risk pregnancy was defined as a history of premature udder development/lactation, an increase of CTUP, purulent/serosanguineous vulvar discharge, or the mare’s systemic illness [15,18]. Dystocia was defined as any stage II impediment that could result from maternal or fetal causes or from the fetal membranes [27,28]. Diagnosis of placental insufficiency was performed retrospectively following macroscopic and histopathological examination of the placenta [14,18].

At birth, a complete clinical evaluation, including complete blood count and serum biochemistry (AU 400 analyzer, Olympus/Beckman Coulter, Lismeehan, Ireland), was performed in all foals born from attended parturition. Based on the judgment of the clinician, blood culture and arterial blood gas analysis (from dorsal metatarsal artery by CO-oximetry using the blood gas analyzer ABL 800 FLEX, Radiometer Medical ApS, Copenhagen, Denmark) were performed when indicated. In all sick foals admitted to the hospital after birth, a complete clinical evaluation, including blood culture, complete blood count, serum biochemistry, arterial blood gas analysis and serum IgG determination (by immunoturbidimetric method; DVM Rapid Test II, MAI Animal Health, Elmwood, WI, USA) were performed. The foals were monitored throughout the hospitalization period by a clinical examination performed every 2–6 h, depending on the severity of the existing conditions.

Sick foals were born from attended parturition or referred to the hospital after birth. The inclusion criteria were: (i) age at admission less than 24 h; and (ii) diagnosis of NE requiring level 1–3 of intensive care, based on the classification proposed by Koterba [29]. 

Sick foals were classified as affected by NE based on their history and clinical signs, especially those of neurological dysfunction [30], with the exclusion of other neurological diseases such as meningitis or trauma. Typical historical events included high-risk pregnancy, dystocia, and/or placental insufficiency, and common clinical signs included loss or absence of the suckle reflex, inappropriate teat-seeking behavior, dysphagia, hyperreactivity, and weakness [7]. Foals affected by NE and other diseases (i.e., sepsis, prematurity/dysmaturity, neonatal isoerythrolysis) were excluded from the study. 

The population was then divided into two groups: 13 foals affected by NE born at the Perinatology and Reproduction Unit (group NE), and 20 foals affected by NE hospitalized within 24 h of life (group exNE).

### 2.2. Clinical Data and Sample Collection

The following data were recorded for each mare: breed, age (years), parity, clinical signs, ultrasonographic findings, cervical swab culture, prepartum treatment, gestation length (days), type of parturition (eutocic/dystocic), length of stage II parturition (min), placenta/foal weight ratio (%), macroscopic and histopathological evaluation of the placenta.

The following data were recorded for each foal born from attended parturition: breed, sex, weight (kg), Apgar score at birth [31], blood glucose at birth (mg/dL) (Medisense Optium, Abbott Laboratories Medisense Products, Bedford, MA, USA), umbilical cord vein (UV) and jugular vein (JV) lactate concentration at birth (mmol/L) (Lactate SCOUT+, Leipzig, Germany), hematobiochemical parameters, clinical signs of NE, organ dysfunctions associated with NE, level of intensive care [29], length of hospitalization (days), and outcome. The same data were recorded for the foals hospitalized after birth, except for the Apgar score and UV lactate concentration. 

All biological samples were harvested as part of the clinical monitoring program of the Unit; owners gave written consent to use samples for research.

The amniotic fluid (AF), UV blood, and plasma/serum sampling from the mare’s JV at parturition (TP) and from the foal’s JV at each time point (at birth/admission: T0; at 24 h from birth/admission: T24; at 72 h from birth/admission: T72), as well as the macroscopic and histopathologic evaluation of the placenta, were collected as described in Part 1 of the study [23]. In 5/13 mares of group NE, placenta samples were collected for molecular biology investigations.

### 2.3. Measurement of NGF and VEGF by ELISA

As described in Part 1 of the study [23], dosage of NGF and VEGF levels was performed using commercial ELISA kits (MyBiosource, San Diego, CA, USA), which detect equine NGF and VEGF (Horse NGF/VEGF ELISA; NGF, Cod. MBS040618; VEGF, Cod. MBS035093). According to the kit’s instructions, samples were centrifuged, incubated with primary antibody, and conjugated with microplate, with HRP-conjugate and two chromogens added in sequence. A sequence of four washings was carried out before adding chromogens. Finally, the stop solution was distributed, and the plate read within 15 min at 450 nm.

Quantification was performed by interpolating the optical density values on a linear standard curve using the GraphPad Prism v 6.0. NGF standard curve ranges from 15.6 ng/mL to 500.0 ng/mL, whereas VEGF standard curve ranges from 31.2 pg/mL to 1000.0 pg/mL.

Each clinical group (H, NE and exNE) was represented in each analytical session.

### 2.4. Measurement of Thyroid Hormones

As described in Part 1 of the study [23], basal thyroid hormone levels of total triiodothyronine (TT3) and total thyroxine (TT4) from the mare JV, UV, and the foal JV at T0, T24, and T72 were determined using the Siemen’s Immulite TT3 and TT4 kits (Immulite Canine TT3 and TT4; Siemens, Oakville, ON, Canada) validated for use with equine serum at the Endocrine Laboratory, Prairie Diagnostic Services, Saskatoon, SK [32,33]. The analytical sensitivity was 0.54 nmol/L for the TT3 assay and 0.15 nmol/L for the TT4 assay.

### 2.5. Placental Gene Expression

As described in Part 1 of the study [23], chorion, allantois, and amnion tissues were homogenized, and total RNA isolation was performed using the RNeasy Microarray Tissue Mini Kit (Qiagen, Hilden, Germany; Cod. 73404) by the automated extractor QIAcube Connect (Qiagen).

Total RNA was eluted in RNase Free Water and quantified through a spectrophotometer (Nanodrop 2000, Thermo Scientific, Waltham, MA, USA) measuring absorbance values at 260, 280, and 320 nm. cDNA was produced using the cDNA the iScript™ gDNA Clear cDNA kit (Biorad, Hercules, CA, USA; Cod. 1725035BUN).

A semi-quantitative real-time PCR was performed using the CFX96 real-time PCR system (BioRad, Hercules, CA, USA). The reactions were performed in a 20 µL final volume including SYBR Green qPCR master mix (BioRad, Cod. 1725274), 0.4 µM of forward and reverse primer mix, and nuclease-free water. The no-RT control was processed in parallel with the others and tested by the real-time PCR for every primer pair to check for eventual genomic DNA contamination, and no-template controls were also added for each gene expression analysis. All primers were designed using the Primer Blast software (NCBI, Bethesda, MD, USA) and synthesized by IDT (Coralville, IA, USA). Specific sequences of primers are listed in Table 1. For the semi-quantitative analysis, the expression of the genes was normalized on the housekeeping gene β-actin (ACTB; NM_001081838.1).

The thermal profile of PCR reactions consisted first of a denaturation step (98 °C, 3 min) and 40 cycles of amplification (95 °C for 10 sec and 60 °C for 1 min), followed by the melting curves (55 °C to 95 °C, Δt = 0.5 °C/s).

Primer efficiency values for all primers were 95–100%, therefore the 2^−ΔΔ Ct^ method was used to perform the analysis.

### 2.6. Statistical Analysis

In order to compare the population affected by Neonatal Encephalopathy (NE and exNE groups) with the healthy population (H group), all biomarkers were expressed as a percentage of the mean of the results obtained in the healthy group at each time point and a t-test was performed to reveal significant differences between the groups. When the biomarker levels of the three groups were compared, based on age at T0, only foals in the exNE group with 0–12 h at admission were compared with those in the H and NE groups.

To assess the correlations between the parameters, Pearson or Spearman correlation coefficients were calculated with Gaussian or non-Gaussian distributions, respectively.

NGF and VEGF levels were correlated in foals JV-plasma, between mares and foals and with the data recorded for each mare at TP (age, parity, gestation length, placenta/foal weight ratio) and the following clinical and haematobiochemical parameters, recorded for each foal at T0: weight at birth, UV and JV lactate concentration, serum creatine kinase, total bilirubin, blood urea nitrogen, creatinine, magnesium, and serum amyloid A.

A *p* < 0.05 was considered statistically significant.

All statistical analyses were carried out using commercial software GraphPad Prism version 8.00.

## 3. Results

### 3.1. Population Characterization

The clinical and histopathological data collected from mares hospitalized for attended parturition that delivered foals affected by NE (group NE) are shown in Table 2. Four/thirteen mares had a high-risk pregnancy associated with increased transrectal CTUP (three/four mares) and both increased transrectal CTUP and systemic illness (surgical colic at 282 days of gestation; one/four mares). In 8/13 mares, a diagnosis of placental insufficiency was reached on the basis of macroscopic and histopathological placenta evaluation. Specifically, placental insufficiency was associated with placental villous hypoplasia in 5/8 mares. Macroscopically, an extensive transition area between the normal chorionic surface and the hypoplasic/discolored surface of the chorioallantois was observed. The histological preparation of the chorioallantois stained with hematoxylin–eosin showed severe hypoplasia of the chorionic villi. In 3/8 mares, placental insufficiency was associated with placental edema. Macroscopically, generalized edematous and heavy fetal membranes, with an increased placenta/foal weight ratio, were observed. The histological section of the chorioallantois showed hyperemia and edema of the chorionic connective lamina associated with mild to severe hypoplasia of the chorionic villi. In 4/13 mares, the clinical condition of the neonate was associated with a dystocic parturition and in 1/13 mares the pregnancy, parturition, and placenta evaluation were apparently normal. As shown in the table, 3/13 mares (23%) from group NE were treated for increased transrectal CTUP with a negative cervical swab, using flunixin meglumine, 1.1 mg/kg, iv, q12h, and pentoxifylline, 8.5 mg/kg, po, q12h, until CTUP returned to normal. In addition, 1/10 mares from group NE received treatment for both increased transrectal CTUP with a negative cervical swab and surgical colic (sodium ampicillin, 20 mg/kg, iv, q8h; gentamicin sulfate, 6.6 mg/kg, iv, q24h; flunixin meglumine, 1.1 mg/kg, iv, q12h; pentoxifylline, 8.5 mg/kg, po, q12h; altrenogest, 0.088 mg/kg, po, q24h).

The clinical data collected from 13 affected foals of group NE born from attended parturition are summarized in Table 3. The clinical data collected from 20 affected foals of group exNE hospitalized within 24 h of life, with an average age at admission of 10 h, are summarized in Table 4. Three/thirteen foals (23%) of NE group and three/twenty foals (15%) of exNE group were born from red bag delivery (premature placenta separation). Overall, 10/33 foals (30.3%) required level three of intensive care [29], which is provided to severely affected neonates. Foals were unable to stand and/or unable to nurse from the mare and need round-the-clock care; this level of care usually involves separation of the foal from the dam, oxygen therapy, parenteral nutrition, inotropes/vasopressor therapy, insulin therapy. Based on the level of intensive care and outcome, foals in the exNE group presented a more severe clinical condition.

Overall, these foals developed a wide range of neurological signs in the immediate postpartum period. Among the most frequent clinical signs of NE, the authors documented: depression (20/33 foals, 60.6%), hypoventilation (16/33 foals, 48.5%), lack of suckle reflex (16/33 foals, 48.5%), lack of affinity with the mare (14/33 foals, 42.4%), severe and prolonged dysphagia (11/33 foals, 33.3%), lateral recumbency (10/33 foals, 30.3%), tongue protrusion (7/33 foals, 21.2%), coma (4/33 foals, 12.1%), and hyperexcitability (4/33 foals, 12.1%). These signs were frequently accompanied by gastrointestinal (abdominal distension, ileus, gastric reflux, gastric ulceration, meconium retention; 13/33 foals, 39.4%), respiratory (abnormal respiratory patterns, dyspnea, hypoxemia; 13/33 foals, 39.4%), renal (oliguria; 11/33 foals, 33.3%), cardiovascular (hypotension, dysrhythmia; 10/33 foals, 30.3%), and metabolic (hypo/hyperglycemia; 6/33 foals, 18.2%) dysfunctions.

The results of the complete blood count, serum biochemistry and arterial blood gas analysis, including JV glucose and UV and JV lactate at T0 (measured through rapid methods) performed in both groups are shown in Appendix A [34,35,36,37,38] in the Appendix A.

### 3.2. Biomarkers (NGF, VEGF, TT3 and TT4) in Biological Fluids

The NGF and VEGF plasma levels and the TT3 and TT4 serum levels found in the foal JV at 0, 24, and 72 h from birth, AF, UV, and mare JV at TP in group NE, and in the foal JV at 0, 24, and 72 h from admission in group exNE are shown in Table 5 (a,b,c). In group NE, NGF levels decreased significantly from T0 to T24 (*p* = 0.0447), VEGF levels decreased significantly from T0 to T72 (*p* = 0.0234) and TT4 levels decreased significantly from T0 to T72 (*p* = 0.0007). In group exNE, NGF levels decreased significantly from T0 to T24 (*p* = 0.0304) and TT4 levels decreased significantly from T0 to T24 (*p* = 0.0055) and from T0 to T72 (*p* = 0.0044).

Figure 1 and Figure 2 represent the comparison of biomarkers *versus* healthy group. Data are expressed as a percentage of the mean of the results obtained in the healthy group at each time point and in each biological fluid analyzed. A significant reduction of TT3 levels was observed in both NE (T24, *p* = 0.0066; T72 *p* = 0.0003) and exNE groups (T0, *p* = 0.0082; T24, *p* < 0.0001; T72, *p* < 0.0001), whereas a significant reduction of TT4 levels was observed only in exNE group (T0, *p* = 0.0003; T24, *p* = 0.0010; T72, *p* = 0.0110).

Among the different biological fluids analyzed, no correlation was found in either the NE or exNE group, but in NE group NGF levels were positively correlated with both TT3 (*p* = 0.0475; r = 0.3424) and TT4 levels (*p* = 0.0063; r = 0.4589), as shown in Figure 3.

### 3.3. Equine NGF/VEGF and Clinical Data

No significant correlations were found between NGF levels in the foal’s JV at T0 and the data recorded for the mares at TP and the selected foal’s clinical parameters at T0, but a significant negative correlation was found between VEGF levels in the foal’s JV at T0 and the lactate concentration at T0 (*p* = 0.0500; r = −0.6444), as shown in Figure 4.

### 3.4. Equine NGF, VEGF, and BDNF in the Placenta

In 5/13 mares in the NE group diagnosed with placental insufficiency, the fetal membranes were subjected to molecular biology investigations. Among all the genes analyzed, in the chorion, no significant differences were found between H and NE groups in terms of the expression of trophic factors and their receptors. In the allantois, a decreased NGF expression was observed in NE group (*p* = 0.0033), while no differences in the expression of P75NTR and TRKA receptors were observed. In the amnion, a decreased BDNF expression was observed in NE group (*p* = 0.0498), with no differences in TRKB receptor expression. Only those genes that showed significant differences between the two groups were included in Figure 5.

## 4. Discussion

In the present study, NGF and VEGF levels were measured in the plasma of mares diagnosed with placental insufficiency and/or dystocia, in the amniotic fluid, in the umbilical vein, in the plasma of their foals affected by NE (in the first 72 h of life—group NE), and in foals affected by NE admitted within 24 h after birth (in the first 72 h of hospitalization—group exNE). The trend of serum thyroid hormones (TT3 and TT4) in the first 72 h of life/hospitalization and the gene expression of *NGF*, *VEGF*, *BDNF,* and their receptors (*TRKA*, *p75NTR*, *FLT-1*, *KDR*, *TRKB*) in the fetal membranes were also evaluated.

The first aim of the present study was to clinically characterize a population of foals spontaneously affected by NE. This is a complex disease recognized across different species, characterized by neurologic dysfunction, but often leading to multiorgan dysfunction [39]. Reviews, case reports, and ongoing investigations demonstrate how the understanding of the disease in foals is constantly evolving [6,11,40,41]. Identification of blood markers of brain injury would be crucial for the diagnosis and the prognosis of NE in foals, and the role of trophic factors requires ongoing investigation, since it is likely to be pivotal.

A specific pathogenesis related to hypoxia, ischemia, and asphyxia may not always be recognized in all cases [11,12,42]. In the present study, a history of placental insufficiency based on the histopathological placenta evaluation, dystocic parturition, including premature placenta separation, was evident in most of the cases reported, but not in all, especially in foals hospitalized after birth, where information related to pregnancy and parturition was often lacking. In group NE, in which pregnancies were monitored and parturition attended, the Apgar score provided a semi-quantitative assessment of the severity of signs occurring in response to peripartum asphyxia, and the following score was assigned: <6 (severe asphyxia), 6–8 (mild asphyxia), or 8–10 (normal foals) [31]. Unfortunately, sick foals are usually referred several hours after birth, when the Apgar score can no longer be performed.

In the present study, NE in foals was clinically recognized by many behavioral abnormalities and the most common signs of neurological dysfunction, include alterations in mental status and in respiratory function, changes in muscle tone, changes in responsiveness, vestibular signs, and autonomic disturbances. Behavioral abnormalities included lack of affinity with the mare, disorientation and abnormal udder seeking, lack of suckle reflex and tongue incoordination, and less frequently abnormal vocalization. In many cases, clinical signs of NE followed very predictable patterns; however sometimes foals appeared normal at birth, but developed behavioral or neurological signs a few hours later, as already described [12]. The foals of exNE group presented more clear and severe neurological signs and, generally, more severe clinical conditions. Differently from group NE, foals born from assisted delivery were hospitalized after birth and did not receive prompt intervention and supportive treatment. Furthermore, the clinical signs most frequently associated with NE have been reported in the affected population, devoid of typical concomitant neonatal diseases, and respiratory, gastrointestinal, renal, and cardiovascular dysfunction appear to be the most common.

Laboratory findings in foals with NE are non-specific and often reflect secondary systemic disease or varying degrees of organ dysfunction associated with NE. In both groups, foals could be hypoxic, hypercapnic, had acid-base and electrolyte disorders, azotemia, and poor glucose control. Hypermagnesemia may be the result of severe tissue damage with cell injury or death, and release of the intracellular magnesium, or related to acidosis and hypoxia [8]. Elevated creatinine at birth is commonly observed in foals that have experienced fetal distress or placental dysfunction. The allantoic fluid contains high concentrations of creatinine, and under pathological conditions there is redistribution of fetal fluids to the fetus, resulting in higher blood creatinine levels [43,44]. Foals born from dystocia showed increased creatine kinase activity because they are likely to be affected by muscle damage [45].

Treatment of NE in foals is largely supportive and directed at controlling neurologic dysfunction and addressing associated multiple organ dysfunction. Goals of therapy should be aimed at supporting perfusion and oxygen delivery via fluid therapy, inopressors and oxygen administration, controlling seizures, assessing renal function, supporting metabolic function through careful nutritional management and blood glucose regulation, and preventing sepsis. The prognosis of foals with a unique diagnosis of NE in this study is in line with those proposed previously [10,12,13], with a survival rate of 80%. In the present study, it was not possible to investigate the prognostic value of NGF and VEGF levels at birth/admission, due to the low number of non-surviving foals; therefore, further investigations in larger cohorts are required.

The decreasing trend of TT4 levels recorded in the serum of NE and exNE foals in the first 72 h of life/hospitalization is in agreement with a previous study conducted on foals aged less than 12 h affected by NE [7]. In the present study, serum TT3 levels were lower in both affected foals born at the Unit and those hospitalized after birth compared to healthy subjects, while TT4 levels were lower only in foals hospitalized after birth compared to healthy ones. Perinatal asphyxia could trigger effects on the neonatal HPT axis, causing the lower levels of thyroid hormones found in affected foals in this study, which aggravate both neurological symptoms and multi-organ dysfunction. The decrease in thyroid hormone levels in foals with NE found in this study could be an expression of altered transplacental transport due to a condition of placental insufficiency, or an expression of non-thyroidal illness syndrome (NTIS) developed after birth [46]. This is a well-recognized syndrome described in patients with severe non-thyroidal illness characterized by low T3 associated with normal or low T4 [47]. NTIS probably represents an adaptive response to a systemic illness with a suppressive effect on the HPT axis and a decreased metabolism preventing organ dysfunction or death. Initially, the conversion of T4 to T3 in peripheral tissues decreases, then, as the severity of illness progresses, T4 concentration also decreases, suggesting dysfunctions at the hypothalamic, pituitary, or thyroid gland level [47]. It should be noted that in this study, reduced TT4 levels were observed in group exNE foals hospitalized after birth, which had a more severe clinical condition.

The few studies on NGF which have been performed in the human perinatal period under pathological conditions are mainly related to intrauterine growth restricted (IUGR) fetuses and neonatal plasma levels [48], preeclamptic women [49], and infants born preterm [50]. It has been reported that circulating NGF is significantly lower in IUGR neonates than in appropriate for gestational age ones [48]. Maternal plasma NGF levels in preeclamptic women are lower than those in normotensive ones [49], while maternal and cord plasma NGF levels were reported to be significantly reduced in women who deliver preterm, suggesting a correlation between reduced cord NGF levels and fetal growth, with likely implications for post-natal neurodevelopmental disorders [50]. Otherwise, foals with NE did not differ significantly from healthy control subjects in terms of plasma NGF and VEGF levels, as well as in terms of umbilical cord vein plasma and amniotic fluid levels. Notably, NGF plasma levels are around 1000 times higher in foals compared to human neonates [48]. Under pathological conditions, no correlations were observed between plasma NGF and VEGF levels and the selected mare’s and foal’s clinical parameters, probably due to the low number of samples and high individual variability. However, a negative correlation was found at T0 between VEGF levels in the foal plasma and lactate concentration. On the contrary, an inverse correlation between NGF levels and symptom severity (as assessed by Glasgow Coma Scale score) has been reported in human children with traumatic brain injury [51]. The pathophysiologic principles of CNS ischemia and hypoxia are shared by animals and humans. Compared to human infants with hypoxic ischemic encephalopathy, foals appear to respond rapidly or more successfully to hypoxia, also in view of the better prognosis of the disease in the newborn foal [12]. It can be postulated that high endogenous levels of NGF may exert a default neuroprotective role in foals.

Although hyperlactatemia does not provide diagnostic information, it does indicate the severity of the disease and the need for early and aggressive intervention or closer monitoring, as it occurs during hypoxia and poor tissue perfusion [52]. VEGF can stimulate angiogenesis and modulate vascular permeability [20], and it is also involved in mediating inflammatory responses [21,22]. Although speculative conclusions on this correlation need to be further evaluated, it cannot be excluded that in affected foals, VEGF is able to restore normal tissue perfusion and oxygenation, leading to a decrease in blood lactate concentrations. Differently from that observed in the healthy population, in NE and exNE foals, NGF plasma levels decreased significantly only in the first 24 h of life/hospitalization, whereas VEGF plasma levels decreased only in group NE. Starting from comparable plasma values at birth, in healthy foals the levels of the two trophic factors drop more markedly than in affected foals. Another interesting result was the lack of the positive correlation that exists in healthy foals between plasma levels of NGF and VEGF at each time point. These findings are not diagnostic, but it cannot be ruled out that a failure to decrease circulating trophic factors in the first 72 h of life may be related to the extent of brain damage. Increased segregation of NGF in the brain of these foals cannot be excluded, and the authors speculate that this change primarily reflects the brain compartment, as suggested by studies conducted in human perinatology, showing that blood levels of neurotrophins are similar to those in the brain [48], changing in infants with neurological disorders due to clinical states of prolonged perinatal hypoxia [49]. In vitro and in vivo animal models have also shown that hypoxia-induced cell death is preceded by a period of NGF up-regulation, suggesting that NGF plays a protective role [53]. In the same direction, clinical studies conducted on infants with hypoxic-ischemic brain injury and treated with intraventricular NGF infusion showed a significant clinical improvement in their neurological condition [54,55], and increased NGF and VEGF segregation in asphyxiated foals may protect neurons against protracted injury.

The second interesting aspect was the correlation that exists in group NE foals between NGF and TT3-TT4 levels at each time point. The close relationship between the trophic factor and thyroid hormones has not been found in healthy foals [23], but experimental studies have long confirmed that thyroid hormones mediate direct effects on NGF-induced expression in neonatal mice [56], and also modulate NGF expression in the cerebellum of perinatal rats [57]. The significance of this positive correlation should be investigated in light of NGF and thyroid-hormones roles in the brain compartment. From a translational point of view, several studies suggest the possible interaction of thyroid hormones with members of the NGF family of neurotrophins and their functional receptors, not only during brain maturation, but also during its maintenance. Thyroid hormones are known to regulate endogenous NGF synthesis under physiological conditions [58,59] and their administration promotes NGF synthesis and increased NGF content in the brain, depending on brain region and post-natal age [60,61,62]. Although further studies are needed to investigate the functional consequences of the interaction between thyroid hormones and NGF during NE, the increased NGF segregation and the relationship observed with TT3-TT4 could indicate a neuronal distress and the need for protection from ischemic damage.

In the NE group of mares, the histological preparation of the chorioallantois stained with hematoxylin–eosin showed varying degrees of hyperemia and edema of the connective lamina associated with mild to severe hypoplasia of the chorionic villi. These findings may indicate an inability of the placenta to meet the increasing metabolic demands of the fetus as pregnancy progresses. Conditions affecting the utero-placental unit, such as placental hypoplasia and placental edema, can cause a decrease in the nutrients and oxygen supplied to both the fetus and placenta, and any deficiency in placental structure and function may be reflected in a corresponding deficit in fetal growth and maturity, leading to a manifestation of NE in the neonate [44]. Despite the low number of samples, the reduced expression of NGF in the allantois and the reduced expression of BDNF in the amnion seems to characterize the fetal membranes of mares with placental insufficiency that delivered foals affected by NE. Neurotrophins are defined as “angioneurins” as they also regulate angiogenesis in the placenta [63] by contributing to the maintenance, survival, and function of endothelial cells [64]. The allantois represents the essential structural framework for an incredibly dense mass of fetal capillaries supported by minimal amounts of allantoic mesoderm [65], whereas the amnion consists of a cuboid epithelial layer that comes into direct contact with AF [66]. Neurotrophins such as NGF and BDNF play an important role in placental development and maturation, acting through autocrine–paracrine mechanisms [67,68]. In women, NGF is reported to be synthetized in the placenta [69], while a recent study also suggests that optimal NGF expression at the feto-maternal interface is essential for a healthy pregnancy [70], influencing the process of angiogenesis [63]. In the present study, the reduced expression of the two neurotrophins could be related to dysregulation of adhesion [71], angiogenesis [72], apoptosis [73], and proliferation [74] pathways. It can therefore be reasonably postulated that, in the mare and in the fetus, decreased expression of *NGF* and *BDNF* in placental tissues may imply the development of impaired maternal and fetal vascular and nervous networks for the interchange of nutrients and stimuli, leading to a clinical manifestation of NE in the neonate.

The main limitation of the study design is that the population of foals affected by NE was not perfectly homogeneous. In some cases, the disease was associated with a “chronic hypoxia” due to placental insufficiency, whereas in others it was associated with an “acute hypoxia” due to dystocic parturition. Nevertheless, the population offered a pure and spontaneous model of equine NE, devoid of typical concomitant diseases such as bacteremia, local infections, sepsis, or prematurity/dysmaturity.

## 5. Conclusions

The present study provides limited but novel information on the role of trophic factors in the early stages of equine neonatal life, when the diagnosis of NE is reached through careful clinical examination. Foals with NE do not differ significantly from healthy control subjects in terms of plasma NGF and VEGF levels, although the levels of the two trophic factors decrease less markedly than in healthy foals. A positive correlation between NGF and both thyroid hormones appears to characterize the first 72 h of life in affected foals, as their fetal membranes seem to be characterized by the dysregulation of NGF and BDNF expression.

Overall, these results provide a starting point for a better understanding of the role of NGF and VEGF during the equine perinatal period under pathological conditions, although they require confirmation by further studies on a wider population of affected foals.

Nevertheless, no blood biomarkers are yet in current clinical use for foals with NE. In the authors’ opinion, the ideal biomarker for identifying equine NE would be stable, measurable during the first hours of life with a high-sensitivity technique, and in an easy-to-access biological sample. Although NGF and VEGF were unable to detect the disease in the population examined, they appear to be promising candidates, which warrants investigation in wider cohorts.

## Figures and Tables

**Figure 1 vetsci-09-00459-f001:**
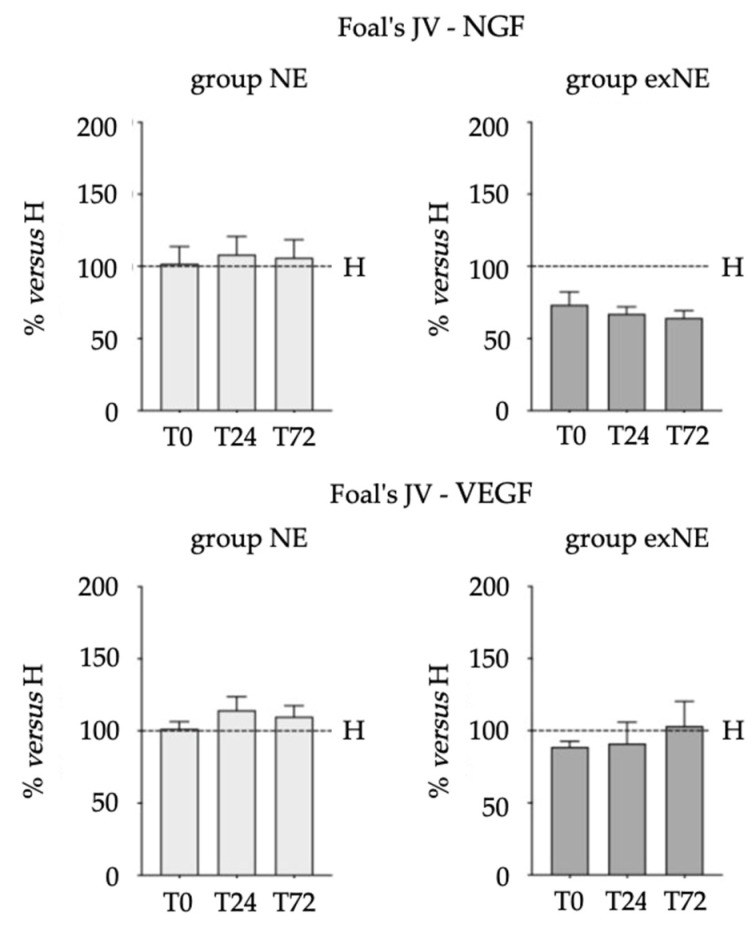
Biomarkers comparison between NE and exNE groups *versus* H group. Levels in samples obtained from foal’s jugular vein at three consecutive time points (T0: birth/admission; T24: h from birth/admission; T72: h from birth/admission) are expressed as percentage of the mean of the results obtained in the healthy group at each time point. ** p <* 0.05; ** *p* < 0.01; *** *p* < 0.005; **** *p* < 0.001.

**Figure 2 vetsci-09-00459-f002:**
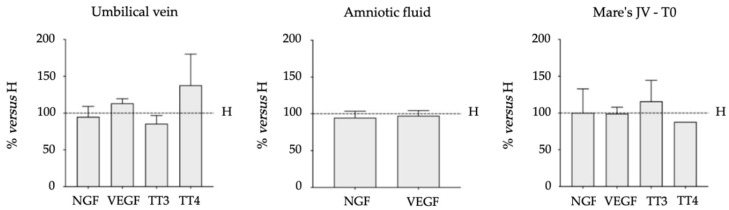
Biomarkers comparison between NE group *versus* H group. Levels in samples obtained from umbilical cord vein, amniotic fluid and mare’s jugular vein (JV) at parturition (TP) are expressed as percentage of the mean of the results obtained in the healthy group (H) at each time point. No differences were observed between groups.

**Figure 3 vetsci-09-00459-f003:**
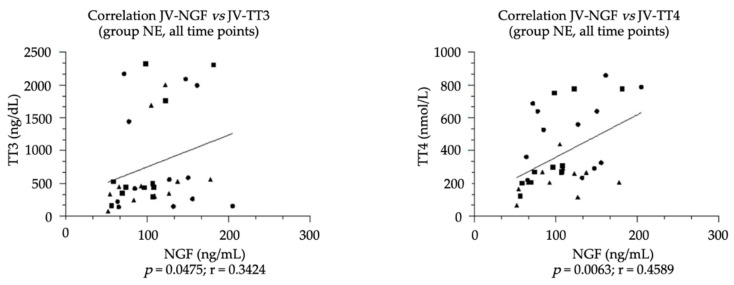
Correlation found in NE group between NGF plasma levels and TT3-TT4 serum levels in samples obtained from foal’s jugular vein at three consecutive time points (T0: birth; T24: h from birth; T72: h from birth). The correlation between NGF and TT3 levels was calculated using Spearman’s non-parametric correlation, whereas the correlation between NGF and TT4 levels was calculated using Pearson’s parametric correlation.

**Figure 4 vetsci-09-00459-f004:**
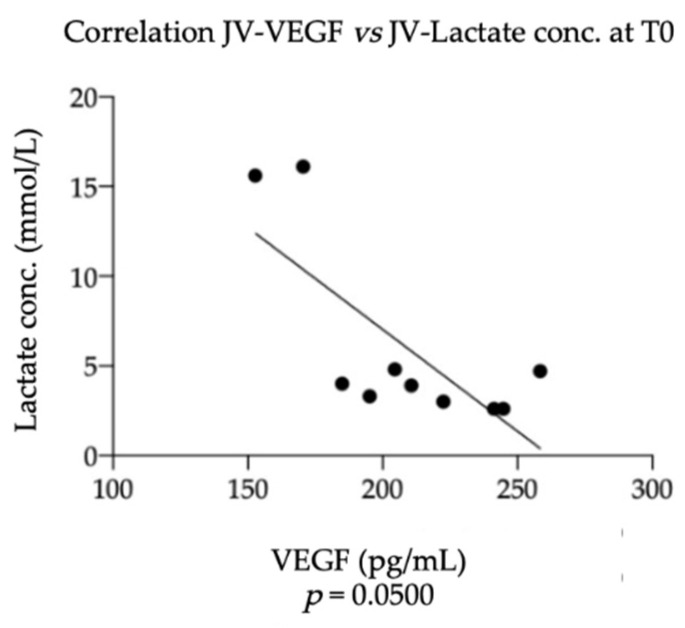
Significant correlation found between Vascular endothelial growth factor (VEGF) levels in plasma samples obtained from foal’s jugular vein and the lactate concentration at T0 (*p* = 0.0500; r = −0.6444). Statistical analysis: nonparametric Spearman correlation.

**Figure 5 vetsci-09-00459-f005:**
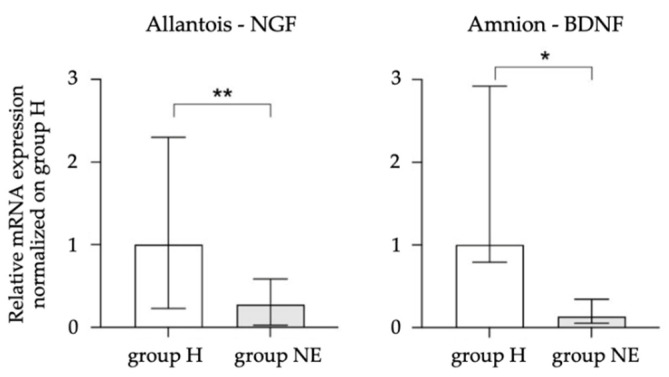
Differences in nerve growth factor (*NGF*) and brain-derived neurotrophic factor (*BDNF*) relative mRNA expression normalized on healthy foals between H and NE groups in allantois and amnion, respectively. * *p* < 0.05; ** *p* < 0.01.

**Table 1 vetsci-09-00459-t001:** List of the gene specific primer sequences.

Genes	Primer Sequences (5′ -> 3′)
*NGF* *Nerve Growth Factor*	Forward: GGGCCCATTAACGGCTTTTC Reverse: CATTGCTCTCTGTGTGGGGT
*P75NTR* *p75 Neurotrophin Receptor*	Forward: GAGCCAACCAGACTGTGTGT Reverse: GGTAGTAGCCATAGGCGCAG
*TRKA* *Tropomyosin Receptor Kinase A*	Forward: GGAGCTGAGAAACCTCACCAT Reverse: GCACAGGAACAGTCCAGAGG
VEGF *Vascular Endothlial Growth Factor*	Forward: AACGACGAGGGCCTAGAGT Reverse: CAAGGCCCACAGGGATTTTCT
*KDR* *Kinase Insert Domain Receptor*	Forward: GATGACAACCAGACGGACAGT Reverse: TTTTGCTGGGCATCAGTCCA
*FLT1* *Fms Related Receptor Tyrosine Kinase 1*	Forward: CTGGCATCCCTGTAACCACA Reverse: AGGGTGCTAGCCGTCTTATTC
*BDNF* *Brain Derived Neurotrophic Factor*	Forward: CATGTCTATGAGGGTCCGGC Reverse: CATGTCCACTGCCGTCTTCT
*TRKB* *Tropomyosin Receptor Kinase B*	Forward: CGGGAACACCTCTCGGTCTA Reverse: CTGGACCAACACCTTGTCTTGA
*ACTB* *Beta Actin*	Forward: TCCCCCCTGGAGAAGAGCTACGAG Reverse: TTTCGTGGATGCCACAGGAT

**Table 2 vetsci-09-00459-t002:** Clinical and histopathological data collected from the 13 mares hospitalized for attended parturition (group NE). SB = Standardbred; QH = Quarter Horse; WB = Warmblood; SD = Saddlebred; US = ultrasonographic; CTUP = transrectal combined thickness of the uterus and placenta; NA = data not available. Data are expressed as mean ±standard deviation (min-max).

Breed	Age (Years)	Parity	US Findings	Cervical Swab	Prepartum Treatment (Y/N)	Gest. Length (Days)	Dystocia (Y/N)	Stage II Labor Length (min)	Placenta/Foal Weight Ratio (%)	Macroscopic Placenta Evaluation	Histopathologic Placenta Evaluation	Mare’s Diagnosis
SB (*n* = 10) QH (*n* = 1) WB (*n* = 1) SD (*n* = 1)	10 ± 3.7 (4–16)	2 ± 1.9 (1–7)	Normal (*n* = 9) Increased CTUP (*n* = 4)	Neg (*n* = 4) NA (*n* = 9)	Y (*n* = 4) N (*n* = 9)	337 ± 10.3 (326–359)	Y (*n* = 5) N (*n* = 8)	19 ± 13.5 (7–55)	11.1 ± 2.4 (8.2–16.7)	Villous hypoplasia (*n* = 4) Extensive edema (*n* = 3) Meconium-stained amnion (*n* = 1) Normal (*n* = 5)	Chorionic epithelium hypoplasia (*n* = 5) Severe edema (*n* = 3) Normal (*n* = 5)	Placental insufficiency (*n* = 8) Dystocia (*n* = 4) Apparently normal (*n* = 1)

**Table 3 vetsci-09-00459-t003:** Clinical data collected from 13 foals affected by Neonatal Encephalopathy born from attended parturition (group NE). SB = Standardbred; QH = Quarter Horse; WB = Warmblood; SD = Saddlebred; Sv = survived to hospital discharge; NSv = not survived; *n* = number of animals. Data are expressed as mean ±standard deviation (min-max).

Breed	Sex	Weight (kg)	Apgar Score (0–10)	Clinical Signs of NE	Organ Dysfunctions Associated with NE	Care Level (1–3)	Hosp. Length (Days)	Outcome
SB (*n* = 10) QH (*n* = 1) WB (*n* = 1) SD (*n* = 1)	Males (*n* = 10) Females (*n* = 3)	45 ± 9.7 (17–55)	7 ± 2.4 (1–10)	Depression (*n* = 7); severe and prolonged dysphagia (*n* = 7); hypoventilation (*n* = 5); lateral recumbency (*n* = 4); lack of affinity with the mare (*n* = 4), lack of suckle reflex (*n* = 4); hypothermia (*n* = 2); weakness (*n* = 2); disorientation (*n* = 1); tongue protrusion (*n* = 1)	Cardiovascular (*n* = 1) Respiratory (*n* = 2) Gastrointestinal (*n* = 5) Renal (*n* = 2) Metabolic (*n* = 2) None (*n* = 6)	Level 1 (*n* = 7) Level 2 (*n* = 4) Level 3 (*n* = 2)	12 ± 8.6 (4–30)	Sv (*n* = 12) NSv (*n* = 1)

**Table 4 vetsci-09-00459-t004:** Clinical data collected from 20 foals affected by Neonatal Encephalopathy hospitalized within 24 h after birth (group exNE). SB = Standardbred; QH = Quarter Horse; SD = Saddlebred; AH = Arabian Horse; Sv = survived to hospital discharge; NSv = not survived; NA = data not available; *n* = number of animals. Data are expressed as mean ± standard deviation (min-max).

Breed	Sex	Age at Adm. (Hours)	Weight (kg)	Mare’s History				Clinical Signs of NE	Organ Dysfunctions Associated with NE	Care Level (1–3)	Hosp. Length (Days)	Outcome
				Age (Years)	Parity	Gest Length (Days)	Other					
SB (*n* = 10) QH (*n* = 6) SD (*n* = 3) AH (*n* = 1)	Males (*n* = 12) Females (*n* = 8)	10 ± 6.4 (1–24)	42 ± 8 (17–56)	12 ± 5.2 (4–21)	3 ± 1.7 (1–6)	334 ± 12.5 (320–357)	Dystocia (*n* = 9) NA (*n* = 9) Placental insuff. (*n* = 2)	Depression (*n* = 13); lack of suckle reflex (*n* = 12); hypoventilation (*n* = 11); lack of affinity with the mare (*n* = 10); lateral recumbency (*n* = 6); tongue protrusion (*n* = 6); coma (*n* = 4); severe and prolonged dysphagia (*n* = 4); hyperexcitability (*n* = 4); head pressing (*n* = 3); convulsions (*n* = 2); barking (*n* = 2); chewing movements (*n* = 2); wandering (*n* = 2); hypothermia (*n* = 1); weakness (*n* = 1); facial hemiplegia (*n* = 1); sialorrhea (*n* = 1); tremors (*n* = 1); anisocoria (*n* = 1)	Cardiovascular (*n* = 9) Respiratory (*n* = 11) Gastrointestinal (*n* = 8) Renal (*n* = 9) Metabolic (*n* = 4) None (*n* = 5)	Level 1 (*n* = 2) Level 2 (*n* = 10) Level 3 (*n* = 8)	11 ± 5 (4–20)	Sv (*n* = 15) NSv (*n* = 5)

**Table 5 vetsci-09-00459-t005:** Biomarker levels in: **a**; group NE—samples obtained from foal’s jugular vein at three consecutive time points (T0: birth; T24: h from birth; T72: h from birth); **b**; group NE—samples obtained from amniotic fluid, umbilical cord vein and mare’s jugular vein at parturition (TP); **c**; group exNE—samples obtained from foal’s jugular vein at three consecutive time points (T0: admission; T24: h from admission; T72: h from admission). Data are expressed as mean ±standard deviation (min-max). *n* = number of samples analyzed; NA = data not available. * Asterisks indicate significant differences from T0, specific adjusted *p* values are listed below: *^1^
*p* = 0.0447; *^2^
*p* = 0.0234; ***^3^
*p* = 0.0007; *^4^
*p* = 0.0304; **^5^
*p* = 0.0055; **^6^
*p* = 0.0044.

a. NE	T0	T24	T72
NGF (ng/mL)	114.9 ± 47.9 (54.6–204.8) (*n* = 13)	93.0 ± 37.7 *^1^ (39.8–181.2) (*n* = 12)	97.5 ± 39.8 (46.5–177.4) (*n* = 12)
VEGF (pg/mL)	199.3 ± 34.9 (150.2–258.4) (*n* = 13)	188.6 ± 54.8 (92.1–261.8) (*n* = 12)	179.8 ± 43.7 *^2^ (95.9–231.1) (*n* = 12)
TT3 (ng/dL)	850.0 ± 826.4 (140.0–2172.0) (*n* = 12)	867.5 ± 830.9 (162.0–2324.0) (*n* = 11)	638.7 ± 618.8 (78.6–2008.0) (*n* = 11)
TT4 (nmol/L)	511.4 ± 220.0 (220.0–860.0) (*n* = 12)	386.0 ± 250.8 (123.0–776.0) (*n* = 11)	227.1 ± 98.5 ***^3^ (66.0–440.0) (*n* = 11)
**b. NE**	**Amniotic fluid**	**Umbilical cord vein**	**Mare’s jugular vein (TP)**
NGF (ng/mL)	139.4 ± 40.2 (61.2–201.9) (*n* = 10)	127.5 ± 57.0 (65.5–214.5) (*n* = 9)	102.2 ± 74.4 (43.5–193.4) (*n* = 5)
VEGF (pg/mL)	264.0 ± 48.4 (138.9–328.4) (*n* = 10)	218.1 ± 33.4 (151.5–261.8) (*n* = 8)	132.8 ± 26.5 (106.0–161.6) (*n* = 5)
TT3 (ng/dL)	NA	350.3 ± 145.0 (110.0–540.0) (*n* = 10)	69.3 ± 34.1 (40.0–118.0) (*n* = 4)
TT4 (nmol/L)	NA	501.4 ± 220.2 (221.0–800.0) (*n* = 10)	18.7 ± 11.5 (12.9–36.0) (*n* = 4)
**c. exNE**	**T0**	**T24**	**T72**
NGF (ng/mL)	82.6 ± 36.8 (56.8–176.9) (*n* = 13)	57.6 ± 14.6 *^4^ (34.6–77.7) (*n* = 12)	59.0 ± 14.0 (42.8–84.7) (n = 9)
VEGF (pg/mL)	174.6 ± 15.0 (165.4–197.0) (*n* = 4)	150.6 ± 48.3 (88.3–204.6) (*n* = 4)	169.2 ± 56.4 (89.6–222.2) (*n* = 4)
TT3 (ng/dL)	368.1 ± 470.0 (60.3–1856.0) (*n* = 13)	279.8 ± 151.3 (99.7–552.0) (*n* = 13)	279.0 ± 111.4 (157.0–505.0) (*n* = 10)
TT4 (nmol/L)	302.5 ± 136.7 (57.8–500.0) (*n* = 13)	196.1 ± 75.2 **^5^ (56.1–292.0) (*n* = 13)	142.0 ± 65.8 **^6^ (46.6–282.0) (*n* = 10)

## Data Availability

Not applicable.

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
