# Peer review of "Study on NGF and VEGF during the Equine Perinatal Period—Part 2: Foals Affected by Neonatal Encephalopathy"

_vetsci, 2022, doi:10.3390/vetsci9090459_

Round 1
Reviewer 1 Report
In the present study, authors presented a comparison between the data collected from first study in normal foals with foals that presented NE. It is well designed and presented. Even though, there are some points to improve.
Abstract
The abstract is too long. The reader will appreciate if it can be simplified in the results section.
M&Ms
Please include the reference of the reagents/products used when it is applicable (ELISA Kits, PCR mix etc).
Line 223 – Please include the table listing the primer sequences.
Lines 223-224 – How were genes normalized?
Results
Line 249 – What causes of high-risk pregnancy?
Lines 251-252 – Is it possible to add illustrative images of placenta?
Lines 247-260 – The authors detailed the clinical findings of the mares used in the study. However, how many mares had no signs of illness until delivery?
Line 318 – Please change Figure to Figures 1 and 2
Discussion
Line 366 – “In the second part of this study, NGF and VEGF levels were measured” This statement gives the impression that 2 studies were made in this manuscript. It may be confusing for a reader that is reading only this manuscript. Please reformulate.
Conclusions
Line 360 Change there to these.
Author Response
REVIEWER 1
In the present study, authors presented a comparison between the data collected from first study in normal foals with foals that presented NE. It is well designed and presented. Even though, there are some points to improve.
The authors thank the reviewer for his careful work and suggestions.
Abstract
The abstract is too long. The reader will appreciate if it can be simplified in the results section.
Thanks for the suggestion. The abstract has been simplified by removing part of the results. The changes are highlighted in the text of the abstract.
M&Ms
Please include the reference of the reagents/products used when it is applicable (ELISA Kits, PCR mix etc).
Line 223 – Please include the table listing the primer sequences.
Lines 223-224 – How were genes normalized?
The authors apologize for the missing information and thanks for the suggestion. We now added in the revised version of the manuscript the product references, the table including the primer sequences, and the normalization method for the gene expression analysis. The new information is highlighted in the text.
Results
Line 249 – What causes of high-risk pregnancy?
Thanks for the suggestion. The causes of high-risk pregnancy have been better specified and highlighted in the text (Results; 3.1. Population characterization).
Lines 251-252 – Is it possible to add illustrative images of placenta?
The authors agree and thank for the suggestion, but the team of pathologists who conducted the histopathological examination of the placenta is not included among the authors and therefore we are not authorized to disclose the images of the fetal membranes.
Lines 247-260 – The authors detailed the clinical findings of the mares used in the study. However, how many mares had no signs of illness until delivery?
Thanks for the suggestion. The causes of high-risk pregnancy have been better specified and highlighted in the text (Results; 3.1. Population characterization). In agreement with the other Reviewer, a brief explanation of the results of histological preparations of the placenta has been added to the results.
Line 318 – Please change Figure to Figures 1 and 2.
Done, thanks for the suggestion.
Discussion
Line 366 – “In the second part of this study, NGF and VEGF levels were measured” This statement gives the impression that 2 studies were made in this manuscript. It may be confusing for a reader that is reading only this manuscript. Please reformulate.
Thanks for the suggestion. The statement has been reformulated and highlighted in the text.
Conclusions
Line 360 Change there to these.
The authors apologize for the mistake and thank for the suggestion. The term has been corrected and highlighted in the text.
Reviewer 2 Report
The manuscript puts a starting point in trying to understand the role of NGF and VEGF in perinatal period of foals, especially considering the appearance of pathologies affecting the nervous system. The data reported in the results and discussed in the conclusions are very interesting even if they require subsequent confirmation, as also indicated by the Authors themselves.
Major point:
-the manuscript is the second part of a study: the first manuscript has been submitted but not yet published. Check if there is any variation in this data
Minor points:
-Abstract: please explain better the "significantly reduction" of TT4 only in the exNE group (lines 36-37), because, as you can see (lines 32-33) also in the NE group "the levels of TT4 decreased significantly"
-paragrapher 2.3: it is necessary to enter the data relating to the reagent used (in particular the antibody used)
-figure 4 is present two times in the text
-the description of evaluation of receptors in "material and methods", "results" and conclusions is missing.
-line 513: the results of histological preparations and stains that have not been described are commented. Please add these data.
Author Response
REVIEWER 2
Comments and Suggestions for Authors
The manuscript puts a starting point in trying to understand the role of NGF and VEGF in perinatal period of foals, especially considering the appearance of pathologies affecting the nervous system. The data reported in the results and discussed in the conclusions are very interesting even if they require subsequent confirmation, as also indicated by the Authors themselves.
The authors thank the reviewer for his careful work and suggestions.
Major point:
-the manuscript is the second part of a study: the first manuscript has been submitted but not yet published. Check if there is any variation in this data.
Thanks for the suggestion. Part 1 of the study (first paper) has been revised by experts in the field and by the editor and no changes have been made to the data for the healthy population. Part 1 of the study is currently in ''final proofreading before publication''.
Minor points:
-Abstract: please explain better the "significantly reduction" of TT4 only in the exNE group (lines 36-37), because, as you can see (lines 32-33) also in the NE group "the levels of TT4 decreased significantly".
Thanks for the suggestion. The authors apologise because they probably needed to express the concept better in the abstract. In both NE and exNE groups, TT4 decreased significantly over time but, when compared to healthy foals, TT4 is lower at each time point only in exNE group, whereas no differences were observed in TT4 levels between NE group and healthy foals.
In agreement with the other Reviewer, the abstract was simplified by removing part of the results, as it was too long and heavy to read. The authors decided to give more attention to the results concerning trophic factors in the abstract, while the data concerning thyroid hormone levels over time are well expressed in the results of the paper.
The changes are highlighted in the text of the abstract.
-paragrapher 2.3: it is necessary to enter the data relating to the reagent used (in particular the antibody used).
Thanks for the suggestion. The revised version of the manuscript now contains all the specific product codes for each reagent used for protein and gene expression quantifications, as highlighted in the text.
-figure 4 is present two times in the text.
Thanks for the suggestion. We apologize for the typo. The last figure describing the gene expression analysis has now been named “Figure 5”, both in the figure legend and in the text.
-the description of evaluation of receptors in "material and methods", "results" and conclusions is missing.
The gene expression analysis was performed on all the genes indicated now in Table 1, as also quantified in the control group in the “Part 1” of the study. However, only the genes showing significant differences between the two groups have been included in Figure 5. An explanation has been introduced in the 'results' section to make the concept clear (3.4. Equine NGF, VEGF and BDNF in the placenta). Thanks for the suggestion.
-line 513: the results of histological preparations and stains that have not been described are commented. Please add these data.
Thanks for the suggestion. All the biological samples (amniotic fluid, umbilical vein blood, mare’s/foal’s jugular vein blood, fetal membranes) were collected as finely described in Part 1 of the study, which is now in ''final proofreading before publication''. Repeating all the sampling procedures in M&M section would make reading the two articles too long and cumbersome. It is a routine histological slide stained with haematoxylin and eosin. The team of pathologists who conducted the histopathological examination of the placenta is not included among the authors and therefore we are not authorized to disclose the images of the fetal membranes. Nevertheless, a brief explanation of the results of histological preparations of the placenta has been added to the “Results” section (3.1. Population characterization) and a comment has been added to the “Discussion” section. Changes are highlighted in the text.
Round 2
Reviewer 2 Report
All the suggestions proposed have been followed by making substantial changes to the manuscript.